# Detect coastal disturbances and climate change effects in coralligenous community through sentinel stations

José C. García-Gómez*⊙, Alexandre R. González⊙, Manuel J. Maestre⊙ ⊙, Free Espinosa⊙

Laboratorio de Biología Marina de la Universidad de Sevilla (LBMUS)/Área de Investigación I+D+i del Acuario de Sevilla/Estación de Biología Marina del Estrecho (Ceuta), Seville, Spain

⊙ These authors contributed equally to this work.
* jcgarcia@us.es

**Data Availability Statement:** All relevant data are within the manuscript and its Supporting Information files.

## Abstract

This study was implemented to assess the Sessile Bioindicators in Permanent Quadrats (SBPQ) underwater environmental alert method. The SBPQ is a non-invasive and low-cost protocol; it uses sessile target species (indicators) to detect environmental alterations (natural or anthropic) at either the local or global (*i.e.*, climate change) scale and the intrusion of invasive species. The SBPQ focuses on the monitoring of preselected sessile and sensitive benthic species associated with rocky coralligenous habitats using permanent quadrats in underwater sentinel stations. The selected target species have been well documented as bioindicators that disappear in the absence of environmental stability. However, whether these species are good indicators of stability or, in contrast, suffer variations in long-term coverage has not been verified. The purpose of this study was to assess the part of the method based on the hypothesis that, over a long temporal series in a highly structured and biodiverse coralligenous assemblage, the cover of sensitive sessile species does not change over time if the environmental stability characterising the habitat is not altered. Over a ten-year period (2005–2014), the sublittoral sessile biota in the Straits of Gibraltar Natural Park on the southern Iberian Peninsula was monitored at a 28 m-deep underwater sentinel stations. Analyses of the coverages of target indicator species (*i.e.*, *Paramuricea clavata* and *Astroides calycularis*) together with other accompanying sessile organisms based on the periodic superimposition of gridded images from horizontal and vertical rocky surfaces allowed us to assess the effectiveness of the method. We conclude that no alterations occurred during the study period; only minimal fluctuations in cover were detected, and the method is reliable for detecting biological changes in ecosystems found in other geographical areas containing the chosen indicator species at similar dominance levels.

## Introduction

The long-term evaluation and quantification of changes in species are crucial for our knowledge of various marine ecosystems [1, 2] and constitute useful tools for the environmental

**Funding:** This research supported by the Regional Activity Centre for Special Protected Areas (RAC/SPA) and the Mediterranean Protected Areas Network (MedPAN) projects, Consejería de Medio Ambiente de la Junta de Andalucía, and co-financed by Autoridad Portuaria de Sevilla, Compañía Española de Petróleos S.A.U. (CEPSA) Foundation and Red Eléctrica de España (REE). Also, research was partially supported by the University of Seville providing salaries to the authors JCG-G and, FE, and Research Foundation of University of Seville (different projects) providing salaries to ARG and MJM. The specific roles of these authors are articulated in the 'author contributions' section. The funders had no role in study design, data collection and analysis, decision to publish, or preparation of the manuscript.

**Competing interests:** This study was partly supported by Autoridad Portuaria de Sevilla, Compañía Española de Petróleos S.A.U. (CEPSA) Foundation and Red Eléctrica de España (REE). There are no patents, products in development or marketed products to declare. This does not alter our adherence to PLOS ONE policies on sharing data and materials.

monitoring, management, and conservation of coastal zones and their associated marine protected areas.

Historically, methodological difficulties related to the monitoring and study of subtidal communities have led to their neglect in many environmental monitoring programs [3], particularly in relation to rocky habitats because of their wide heterogeneities, the fixed nature of the resident sessile organisms, the frequent difficulty of achieving access, and the fact that the sampling must be undertaken through indirect methods from boats. Warwick [4] indicated that the study of these communities on wide spatial scales involves a series of technical and implementation difficulties that are not present for work on soft bottoms, the water column, or the intertidal zone. For these reasons, studies related to the environmental monitoring, surveillance, and impacts of various factors on the sublittoral benthic environment have focused on endofauna (macroinvertebrates) associated with soft substrates [5–12]. Many biotic indices based on these animals have been proposed [13–15], and many of these indices have been used for or adapted to the requirements of the European Community Water Framework Directive [16–18].

Indeed, regarding monitoring methods based on indicators in the scope of the European Community Directives [19–20], the research has centred on macroalgae on hard substrates [21–28] and seagrasses [29–32], whereas macroinvertebrates associated with high-diversity rocky subtidal habitats (which have greater abundances of sensitive indicator species) have been less well studied. However, some studies have sought to establish the ecological statuses of coralligenous assemblages [33–39]. The environmental information that can be provided by these assemblages is very powerful and reliable [40–43]; therefore, this faunal approach to rocky habitats could help to fill an important knowledge gap within the European Community Directives [19–20].

Of particular importance is the information that can be provided by pre-coralligenous and coralligenous rocky bottoms because of their high biodiversities and the abundances of sensitive colonial species with long life cycles. Although there are some long-term studies of species that are representative of coralligenous communities [44–49] short- and medium-term studies are more common [50,51].

In contrast, growing concern about the effects of climate change has led to more studies of epibenthic (rocky bottom) species in relation to coral bleaching [52–54], seagrass meadows [55], invasive alien species [56,57], and even the role of the marine reserve biota in climate change monitoring [58–60]. These studies have also highlighted the severe environmental impacts on benthic organisms that have arisen from abnormal temperature increase events in the western Mediterranean [61–63].

The great fragility of highly structured and mature benthic communities associated with hard substrates, such as those of pre-coralligenous and coralligenous areas, has encouraged non-invasive study methods based on video footage or photos of species that are fixed to the substrate in random quadrats [36,51,64–75]. However, the absence of long-term series that have empirically validated these methodologies as systems for detecting changes in communities has been limiting.

Additionally, previous studies have focused on characterising and/or monitoring benthic communities, which implies greater complexity with respect to implementation mainly due to the high diversity of species present in these enclaves, which, in turn, entails greater effort and difficulty associated with the identification of the taxa. However, the Sessile Bioindicators in Permanent Quadrats (SBPQ) methodology, which was proposed by one of the authors of the present paper [43], differs from other methodologies, mainly in that the quadrats are permanently fixed to the rocky bottom and that the coverage of a previously selected target species must be at least 10% within each quadrat. Fixed quadrats allow changes in the coverage of

target species and the rest of the benthic community to be detected without the need for a high number of replicates, and with the certainty that these differences are not due to the characteristic spatial heterogeneity of this kind of habitat. The simplicity and relatively low cost of the method allows the monitoring of the stations to be repeated in the short term, which is essential in the detection of invasive species or other anthropogenic disturbances that require rapid action. The method is not intended to assess the biodiversity or ecological status of these communities, but to establish an early warning system that allows the detection of changes in the coverage of target species. Based on the detection of these changes, specific studies can be designed to assess the degree and origin of the disturbance.

## Objectives

This study aims to assess the underwater environmental SBPQ alert method, which focuses on monitoring preselected sessile and sensitive benthic species associated with rocky coralligenous habitats using permanent quadrats in underwater sentinel stations. The selected target species have been well documented as bioindicators that disappear in the absence of environmental stability via acute impacts. However, it has not been verified if these species are good indicators of long-term stability. The purpose of the present study was to assess the portion of the method based on the hypothesis that, over a long temporal series (a ten-year period in this study) in a highly structured and biodiverse coralligenous assemblage, the cover of sensitive sessile species (*i.e.*, the target species) does not change over time if the environmental stability of the habitat is not altered.

## Materials and methods

### Study zone

This study was performed on the benthic community of two sentry stations on a rocky bottom. The station is situated within the Strait of Gibraltar Natural Park (southern Spain) inside the Grade A protection zone (*i.e.*, the maximum protection area within the Natural Park) characterized by rocky bottoms with a moderate slope that host a high biodiversity and species richness, dominated by species such as *Eunicella sp.*, *Paramunicea clavata*, *Astroides calycularis*, *Pentapora sp.*, *Crambe crambe* and *Salmacina sp.* in the Punta Carnero locality (Algeciras), but it is proximal to zones under strong anthropic pressure (Fig 1).

Despite its enormous ecological wealth and high level of protection, the SBPQ sentry station is at notable risk of anthropogenic disturbance. This is due to both to its proximity to the particularly industrialised Bay of Algeciras, where many different pollution sources can be found (thermal power plant, chemical industry, petrochemicals, bunkering activities, etc.) [76,77] and the high level of marine traffic through the area. Due to the second factor, the last decade has seen polluting events related to hydrocarbon spills of varying magnitudes including serious spills from the ships *Sierra Nava* and *New Flame*, both of which occurred in 2007 [78].

Data regarding the bottom water temperature in the study area were obtained from data base of the regional environmental authorities [79]. The implementation of the study was notified to the managers of the the Strait of Gibraltar Natural Park and to the competent environmental authority of Andalusia Government (Regional Ministry of Environment and Territory Planning, CMAYOT). In fact, the CMAYOT has funded the publication of the spanish verison book where the methodology SBPQ is included (43). In adition the first author of the current study is part of the governing board of the Strait of Gibraltar Natural Park. No permits were required for this work.

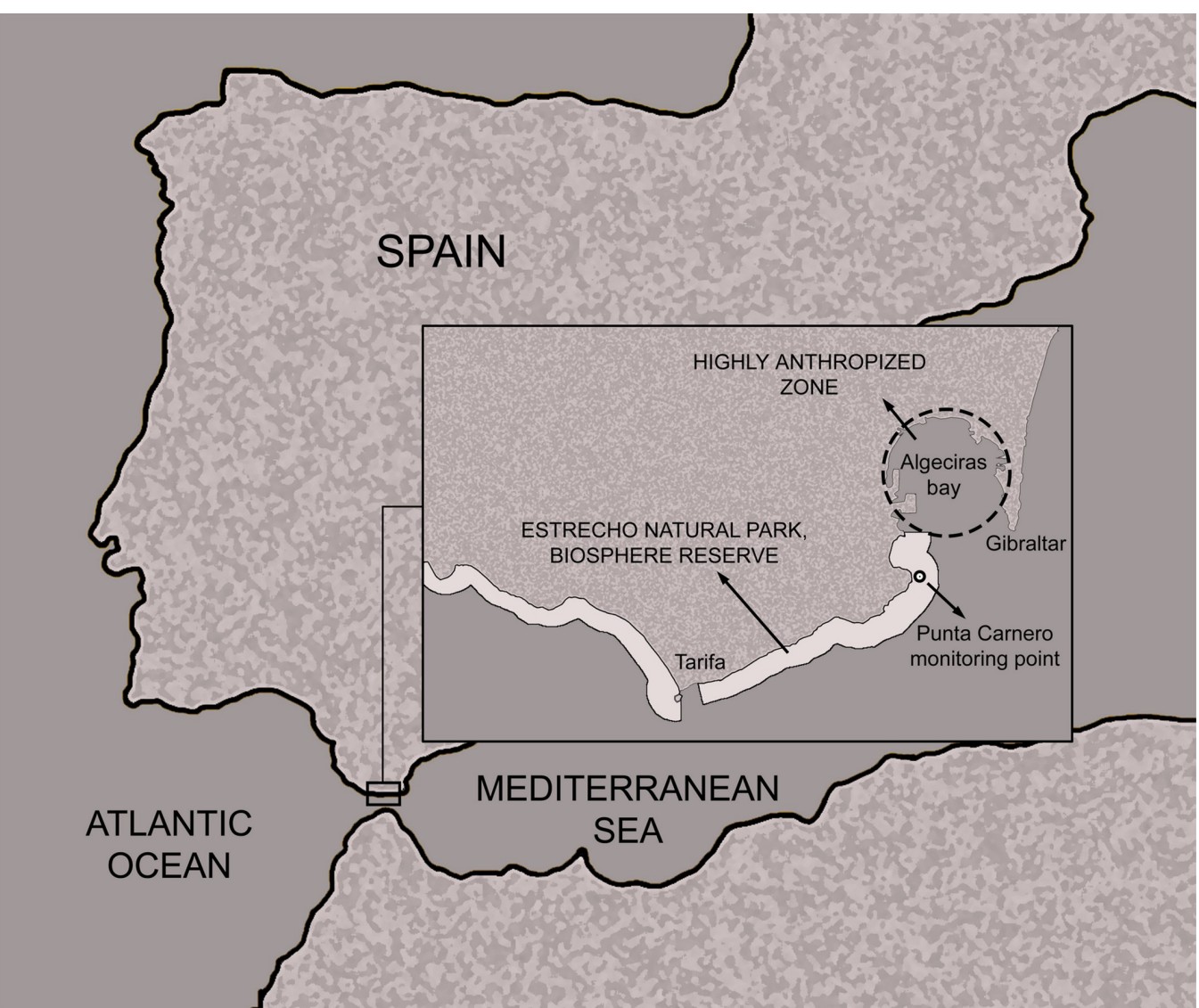

**Fig 1. Location of the sampling point in the Strait of Gibraltar Natural Park.**

## Application of the SBPQ methodology for its validation: Target species, permanent grids, and underwater sentinel stations

The SBPQ method was proposed by García-Gómez [43] together with an identification guide for sensitive indicator species vs. tolerant species, and was published by The Regional Activity Centre for Specially Protected Areas (RAC/SPA). The RAC/SPA was established in Tunis in 1985 and is responsible for assessing the status of the natural heritage and helping Mediterranean countries to implement the Protocol concerning Specially Protected Areas and Biological Diversity in the Mediterranean (SPA/BD Protocol), which came into force in 1999.

This methodology has been designed as a simple, non-invasive, underwater environmental alert tool for the potential early detection of environmental impacts of anthropic origin in the sublittoral system: in the short term (local alterations derived from pollutants from industries or emissions of urban origin, coastal dredging or civil engineering works on the coast, intrusion of exotic

species with invasive potential, among other sources of alteration of coastal waters), and in the medium or long term (global warming). Even though the method is able to detect the presence of invasive species, further studies are required to test the reliability of the method for detecting other potential impacts of anthropic origin. Recently, the SBPQ methodology has already been used successfully in the early detection of exotic species with invasive potential, in particular the invasive seaweed *Rugulopterix okamurae* [80]. The Fig 2 shows a temporal variation of benthic mean percent species coverage at Tarifa Island monitoring by SBPQ method from years 2013 to 2017.

It is focused on the management of sensitive indicator target species (benthic and sessile) vs. tolerant species in the Western Mediterranean and can be used in other parts of the world once the native species have been selected. The SBPQ method has been proposed for widespread use, not only by scientists, but by sports divers linked to diving clubs or centres that could be involved in underwater environmental monitoring of the coastal environment (Citizen Science and "Working with Nature" philosophies).

Synthetic adaptation of the SBPQ protocol

1. Choice of vertical walls of pristine rocky bottoms, preferably between 20 and 35 metres deep, biologically structured and of high biodiversity, with the presence of adult-sized target species, sensitive, benthic and sessile indicators, that are visible underwater, preferably colonial, with a long life cycle, and abundant compared to the local macrobiota.

2. Selective installation of three to five permanent quadrats of 1x1 $m^2$ (not chosen at random), located on patches of at least one previously selected target species that represents at least 10% of the total coverage per quadrat (Fig 3). The method has recently been updated with the objective of minimizing the degree of intrusion on these fragile habitats. For this purpose, a single hole is drilled on the rock and a small plastic bar marks off four 50 x 50 cm detachable monitoring quadrats that are fitted each time monitoring is carried out. This

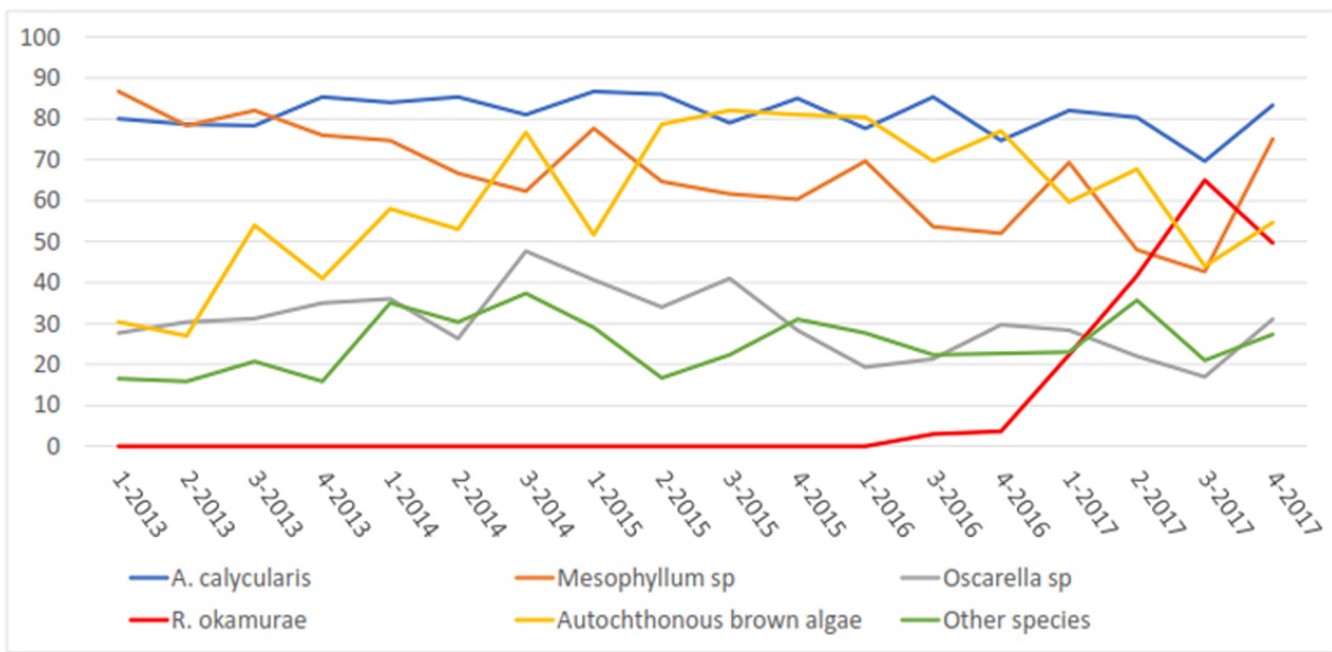

**Fig 2. Temporal variation of benthic mean percent species coverage at Tarifa Island monitoring station fixed quadrats from years 2013 to 2017 (1, 2, 3, 4 refer to random sampling times within each year).** Species with less than 10% coverage (*Alcyonium sp*., *Aplidium sp*., *Asparagopsis armata*, *Crambe sp*, *Ircinia sp*. and *Polycitor adriaticum*) were grouped under 'Other species'. Taken from (García-Gómez et al., 2020).

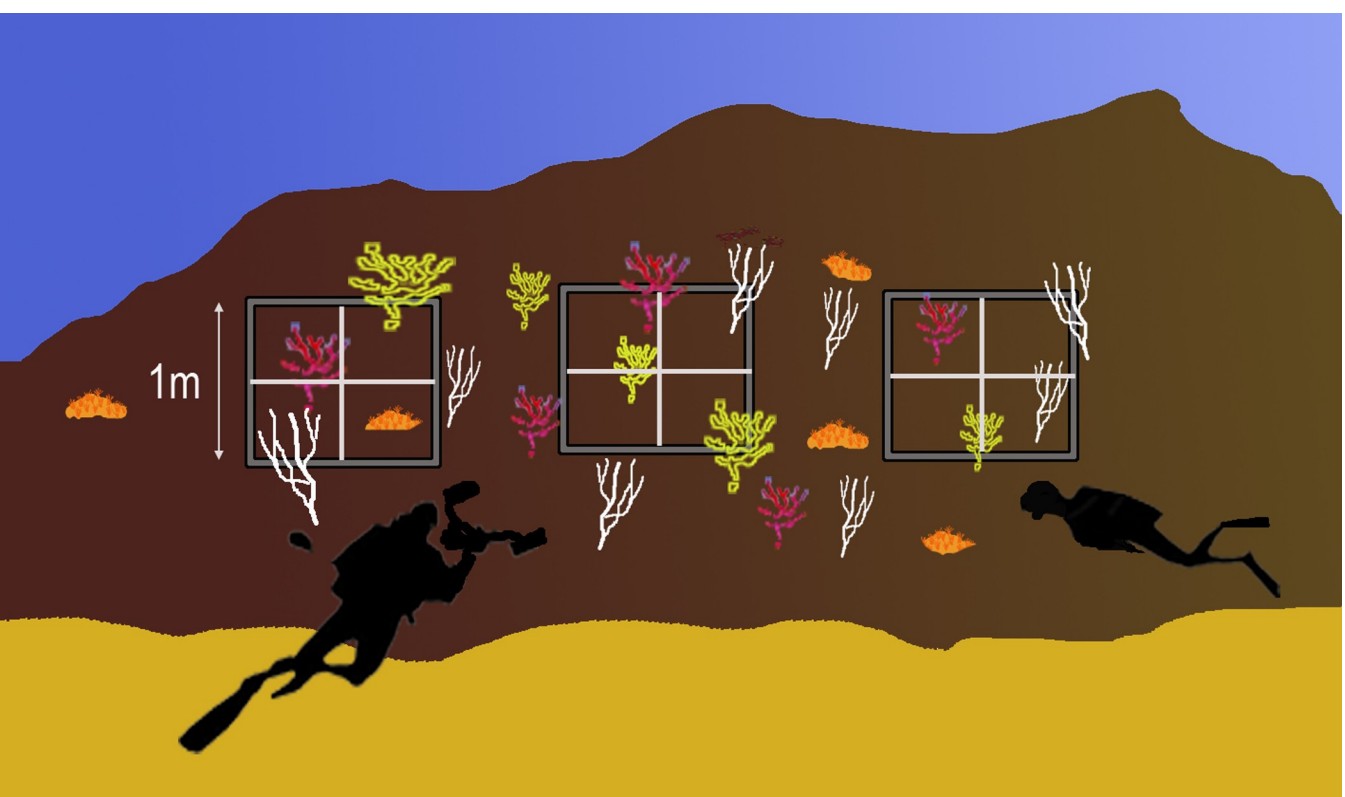

**Fig 3. Sentinel station showing permanent quadrats of 1x1m located on patches of target species.**

avoids the use of permanent fixed quadrats with many screws, therefore reducing effort, cost, maintenance and, most importantly, the impact on communities. As explained above, three to five of such monitoring points would be installed per site (Fig 4). A hand drill (Fig 4A), used in climbing, is proposed in the method. This tool is a cheap and relatively simple way to fix the pieces to the rock and can be supplied to diving clubs that are interested in being part of the monitoring network.

The anthozoans *Astroides calycularis* (Pallas, 1766) and *Paramuricea clavata* (Risso, 1826), which are currently listed as 'vulnerable' by the IUCN, were selected for monitoring. These species are vulnerable to increases in temperature and sensitive to deterioration of the environmental quality of the marine environment. Both species are well referenced in ecological studies of coastal benthos [42,43,63, 81–88]. These species should always be moderate to large (preferably colonial) and highly visible while immersed. Moreover, some studies have indicated that typical coralligenous species, such as *Paramuricea clavata*, exhibit extremely low temporal variability [88]. Additionally, these species are highly representative of the community under study. Therefore, the quantitative stability (cover) of these sensitive species over time in the absence of phenotypic signs of stress (*e.g.*, epibiosis, partial necrosis, bleaching) would allow us to infer that the community has remained healthy. Therefore, focusing the monitoring effort on sensitive species that are representative of the community should allow the appropriate inferences to be drawn.

## Sampling procedure

Four fixed 1-m$^2$ PVC quadrats were installed in two different locations. One setup had a vertical orientation (shady), and the other had a horizontal orientation (sunny) at a depth of 28 m

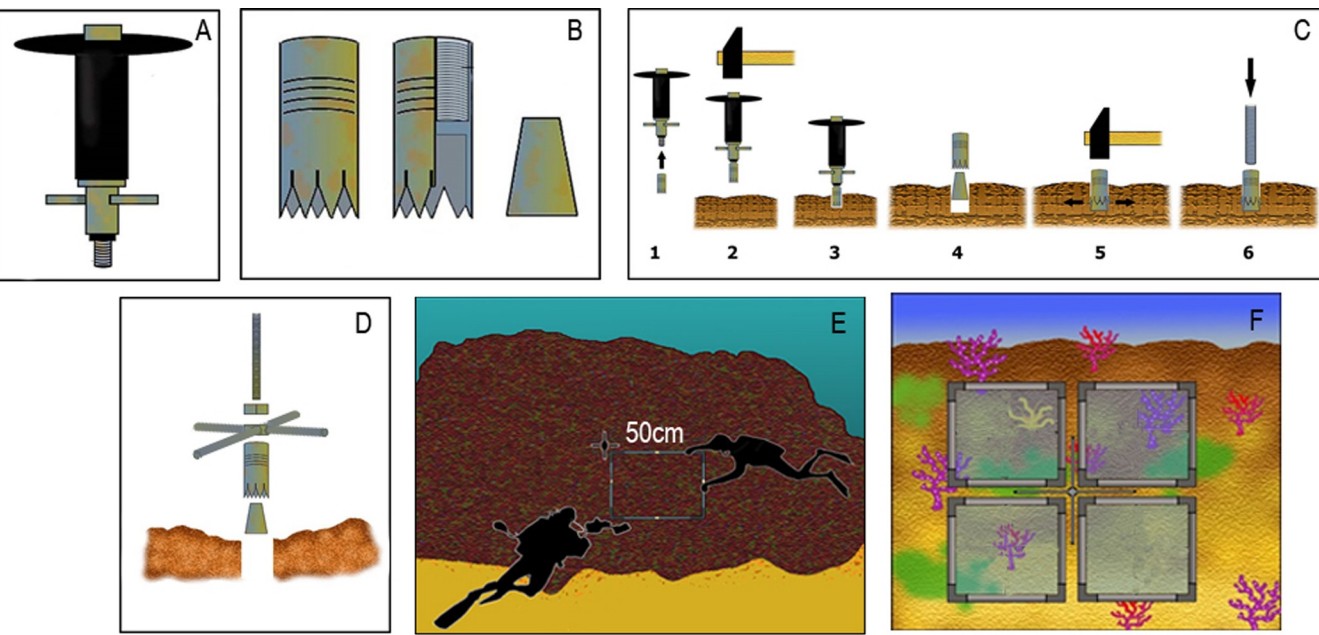

**Fig 4. The updated method with the objective of minimizing the degree of intrusion.** A: Hand drill, B: Expansion anchor bolt, C: Drilling and bolt fixing sequence, D: Metal crosshead fixation, E: Sequential allocation of the detachable quadrat, F: Representation of a complete monitoring point.

(coordinates 36° 4,770 N - 05° 25,184 W; GPS, DATUM WGS 84). These quadrats were placed on surfaces containing target species. The covers were monitored over ten consecutive years (2005–2014) in spring and summer seasons.

A four video footage of 50 x 50 cms were filmed sequentially, covering a total of 1 x 1 m quadrat, in order to gain higher resolution of the subsequent photos taken from the video.

A total of 320 photographs were digitally manipulated using the *Adobe Photoshop* 6.0 (2000) program as follows: four 50x50-cm areas were created for each quadrat (Fig 5A); the contrast and colour saturation were adjusted (Fig 5B); a complete digital frame was created (Fig 5C); and a digital grid that was adjusted to the perimeter of the monitoring area was superimposed (Fig 5D). To correct the angular deformation.

To assess the degree of the cover of the target species, a system involving the determination of the presence/absence of cover using the digital grid was applied. This system aided estimation accuracy [89,90] and optimised the working time [64,91]. Nevertheless, it should be noted that, in multistratified communities, the system tends to underestimate the cover [64,69,89–92] because larger species "mask" those that developed underneath them.

## Statistical analyses

Repeated measures analysis of variance (RM-ANOVA) was applied to test whether the mean coverage of the target species significantly varied either through time (intra-subject factor 'time'; ten levels: 2005 to 2014) or according to orientation (inter-subject factor 'orientation'; two levels: horizontal and vertical). Mauchly's test of sphericity [93] was used to test the assumption that the variances in the differences between all possible pairs of groups were equal. When the sphericity condition was not verified, the F test value was corrected with the Greenhouse-Geisser epsilon index [94,95]. The factors of time and orientation were considered orthogonal and fixed. The same two-factor design was considered to test for any significant differences in the mean coverages via RM-permutational analysis of variance

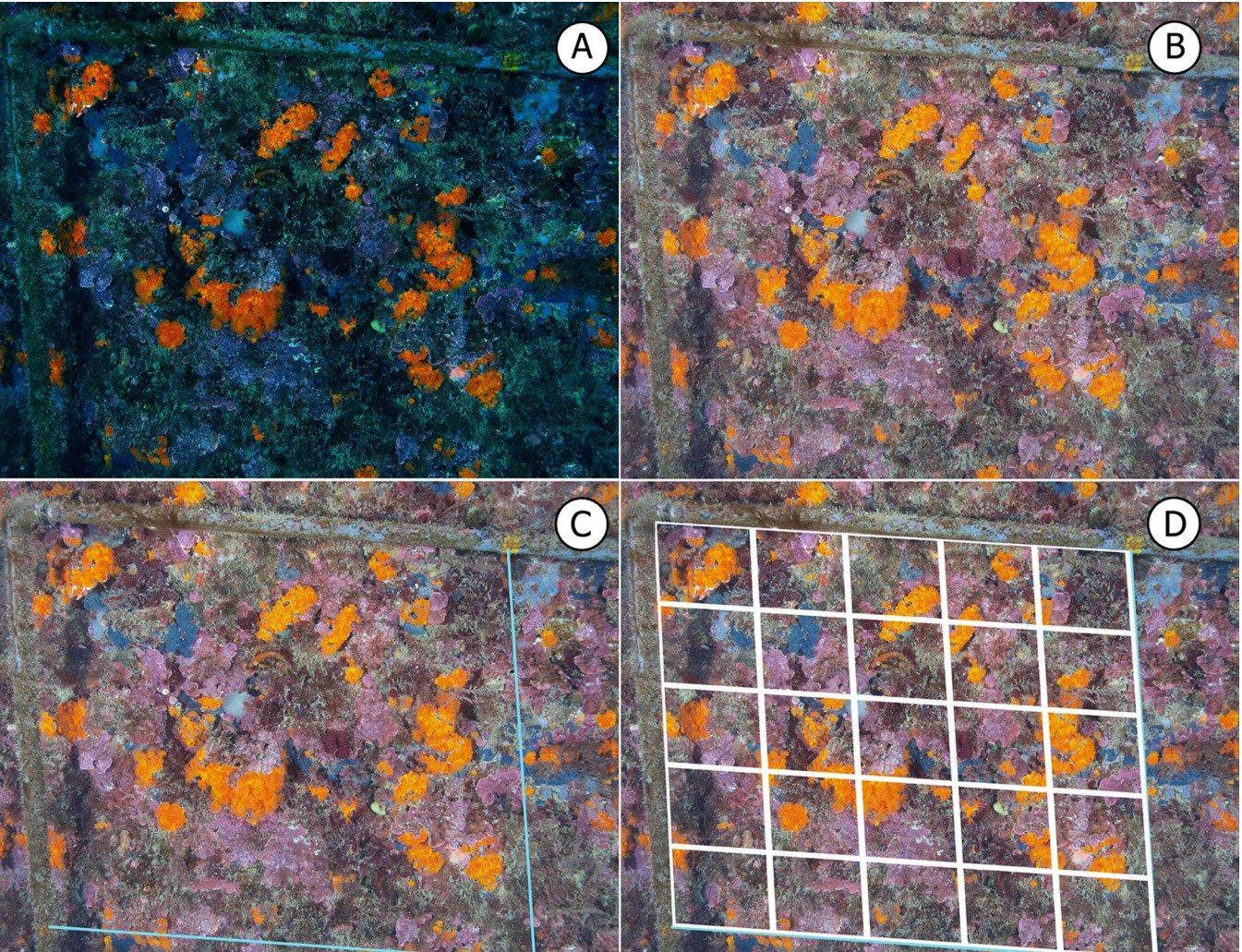

**Fig 5. Digital manipulation of the captured images prior to analysis.**

(RM-PERMANOVA). Univariate analyses were performed using SPSS.13© according to the guidelines of Pardo and Ruiz [96].

For the multivariate analyses, square root transformations were applied to the data, and the analyses were performed using the Bray-Curtis similarity. The Bray-Curtis similarity matrix was used to generate a non-metric multidimensional scaling (nMDS) analysis, and the Kruskal stress coefficient was calculated to test the ordination [97].

The multivariate analyses were performed using the PRIMER6 software (complete with the PERMANOVA+ package) [98–101].

## Results

The average covers for the five most abundant species (including the two pre-selected sensitive target species *A. calycularis* and *P. clavata*) over the period of 2005 to 2014 at the two locations (horizontal and vertical) are presented in Figs 6 and 7.

Fig 8 illustrates the evolution of the covers of the sampled indicator species *A. calycularis* and *P. clavata* at both locations (horizontal and vertical) together with the time series of the

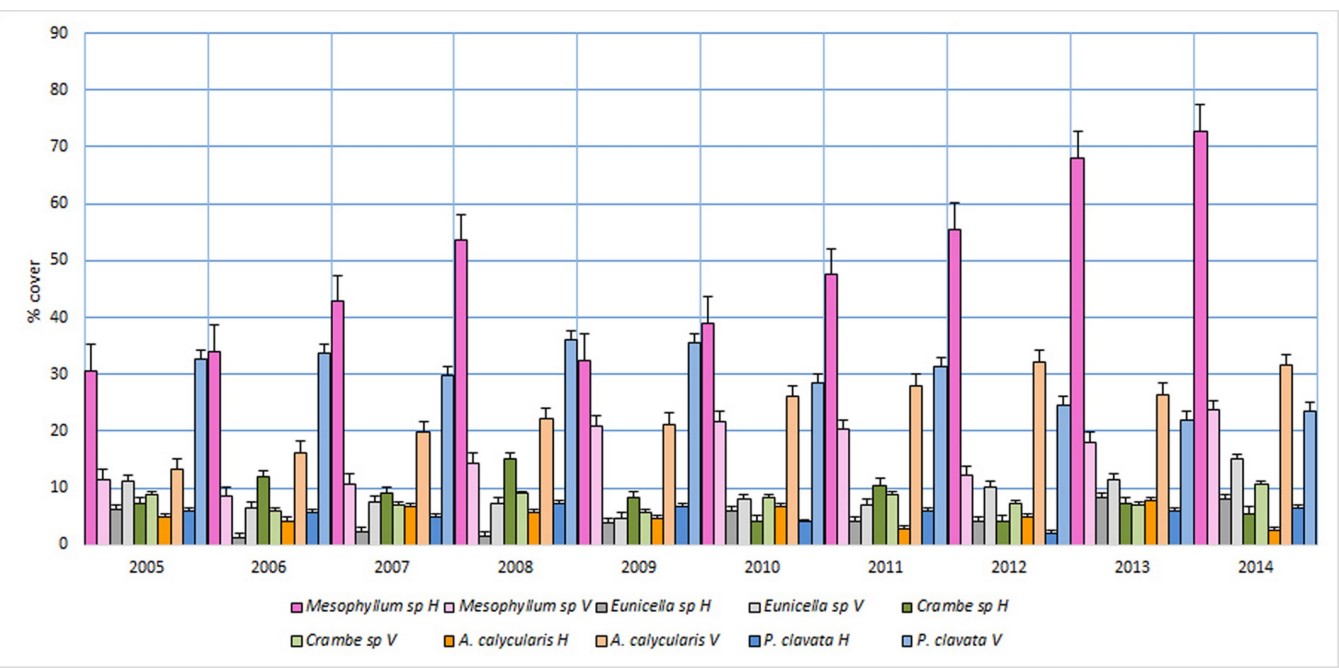

**Fig 6. Graphic representation of the evolution of the cover of the main species observed at the monitoring stations.** H: horizontal; V: vertical.

background water temperatures in the sampling zones. These series reflected no anomalous increases or decreases in temperature and only exhibited oscillations that were attributable to seasonality. The covers of both species were clearly greater in the vertical orientation than in the horizontal orientation. An increase and a decrease were observed in the covers of *A. calycularis* and *P. clavata*, respectively, but there were no significant differences. This decrease in the coverage of P. clavata throughout the period studied coincides with that observed in other studies [47]. Environmental factors such as climate change may be affecting certain species more slowly these trends may be analyzed when they have a longer period.

RM-ANOVA of the cover of each species in each of the two orientations (Table 1) indicated significant differences ($p < 0.05$) in the covers of *Mesophyllum* sp. and *Salmacina incrustans* in the shady (vertical) location over the time period. Regarding the sunny (horizontal) location, the covers of *Mesophyllum* sp. and *Crambe* sp. exhibited significant differences ($p < 0.05$). No significant differences ($p > 0.05$) were detected in the covers of the remaining species.

The RM-PERMANOVA analysis indicated significant differences in the coralligenous assemblages ($p < 0.05$) for the orientation factor (*i.e.*, vertical vs. horizontal) but not for the time factor. The interaction of these two factors was also not significant (Table 2).

The nMDS analysis (Fig 9) revealed a clear differentiation of the values between the vertical and horizontal orientations (Stress: 0.14).

## Discussion

### Sampling procedure and analysis of the submarine images

Previous studies have indicated that fixed monitoring stations constitute one of the most robust methods for detecting changes in benthic communities over time [69,102–105]. An area of at least 1 m$^2$ is sufficient for integrating various colonies and individuals of diverse fauna in a single sample, and such diversity is a typical characteristic of benthic rocky bottom communities [69,106]. However, the definitive sampling grid size must be defined by the final

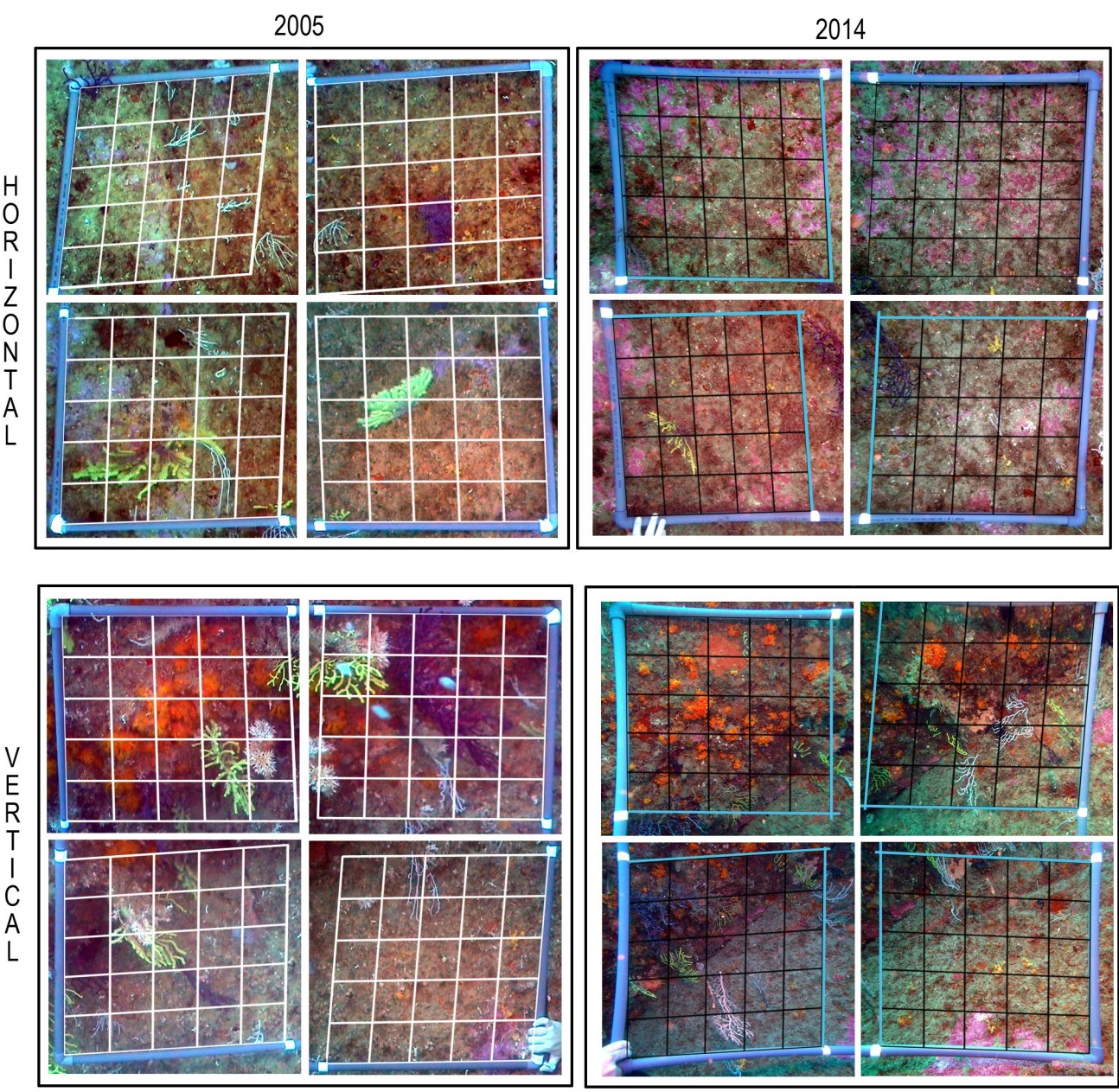

**Fig 7. Photos of the same replicate taken in 2005 and 2014 for both orientations.**

objective of the study [107]. In our case, the objective was not an exhaustive description of the existing community but rather the evaluation of possible changes in the covers of sessile target species within the community. Similarly, the number of quadrats or replicates used must be defined by the target objectives. In other studies of rocky bottoms or surfaces, three [36,108–109], five [110], six [111], and eight replicates [112] have been employed. In all of these studies, the replicates had areas of significantly less than 1 m$^2$ with the exceptions of the studies by Fraschetti *et al*. [109] and Piazzi *et al*. [36] who also used areas of 1 m$^2$.

Image analysis based on a monitoring system comprises a non-invasive method that does not interfere with the natural development or evolution of the studied community [69, 92, 103, 113,

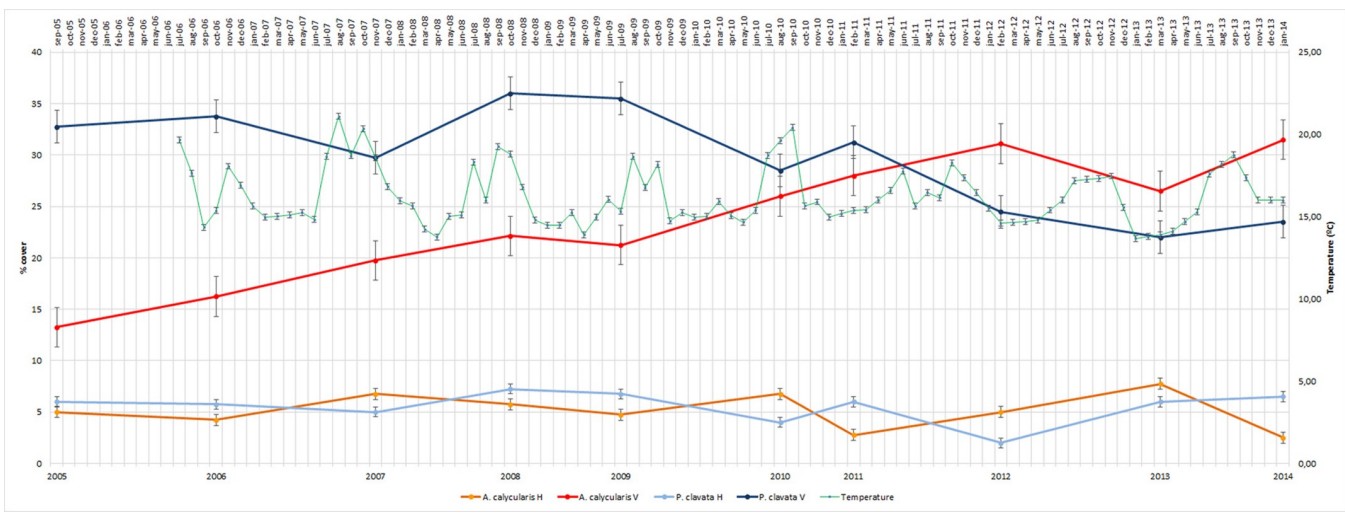

**Fig 8. Graphic representation of the evolution of the cover of the two main indicator species (*A. calycularis* and *P. clavata*), together with the bottom water temperature time series.** The bars indicate the standard deviation. H: horizontal; V: vertical.

114]. This method also allows for rapid data collection and permanent data record generation. The images were captured using videos rather than photographs mainly due to video's greater speed and versatility when obtaining data [115–117]. Video was thus time-saving and consequently allowed for the optimisation of diving operations, which was very important given the depth at which the study was performed and the number of replicates that were monitored.

## Choices of high biodiversity habitats, target species, and other companion organisms

In general, sublittoral marine habitats are characterised by diversity that increases with depth [66, 118, 119]. Deep communities are dominated by animals, are better structured, and exhibit

**Table 1. Results of one-way repeated measures ANOVA on coverage of each species either in vertical and horizontal substrata.**

| | | *A. calycularis* | | *P. clavata* | | *Crambe* sp. | | *Eunicella* sp. | | *Mesophyllum* sp. | | *P. fascialis* | | *S. incrustans* | |
|---|---|---|---|---|---|---|---|---|---|---|---|---|---|---|---|
| **Source of variation** | **df** | **F** | **P*** | **F** | **P*** | **F** | **P*** | **F** | **P*** | **F** | **P*** | **F** | **P*** | **F** | **P*** |
| **Vertical** | | | | | | | | | | | | | | | |
| **Time[+]** | 9 | $1.159^2$ | 0.335 | $1.647^2$ | 0.171 | $0.793^2$ | 0.557 | $0.632^1$ | 0.359 | $3.474^2$ | **0.018** | $2.735^2$ | 0.058 | $2.931^2$ | **0.021** |
| **Residuals** | 10 | | | | | | | | | | | | | | |
| **Mauchly's test of sphericity** | | p<0.001 | | p<0.001 | | p<0.05 | | p = 0.305 | | p<0.01 | | - | | - | |
| **Transformation** | | None | | none | | none | | none | | none | | none | | none | |
| **Horizontal** | | | | | | | | | | | | | | | |
| **Time[+]** | 9 | $0.671^1$ | 0.277 | $0.593^1$ | 0.446 | $5.720^2$ | **0.002** | $1.749^2$ | 0.157 | $0.910^1$ | **0.006** | $1.087^2$ | 0.365 | $0.890^2$ | 0.426 |
| **Residuals** | 10 | | | | | | | | | | | | | | |
| **Mauchly's test of sphericity** | | p = 0.490 | | p = 0.052 | | p<0.01 | | p<0.001 | | p = 0.101 | | - | | - | |
| **Transformation** | | None | | none | | none | | none | | none | | none | | none | |

* Greenhouse-Geisser correction was used when the sphericity condition was not verified.

[+]Time correspond with years from 2005 to 2014.

[1] Pillai´s trace.

[2] Greenhouse-Geisser F.

**Table 2. Results of repeated measures PERMANOVA.**

| Source | df | SS | MS | Pseudo-F | P(perm) | Unique perms |
|---|---|---|---|---|---|---|
| Time | 9 | 2150.5 | 238.95 | 2.4353 | 0.079 | 998 |
| Orientation | 1 | 30338 | 30338 | 309.2 | **0.001** | 998 |
| Ti x Or | 9 | 1207.9 | 134.21 | 1.3678 | 0.359 | 999 |
| Res | 60 | 5887.1 | 98.118 | | | |
| Total | 79 | 39583 | | | | |
| Transformation | | None | | | | |

less abrupt dynamics and smaller temporal changes than shallow communities or those dominated by algae [66]. Additionally, as pointed out by Ballesteros [120], within the scope of marine protected areas (MPAs), the selection of a limited number of representative and/or key species from such communities is a sound strategy for aiding their understanding and management.

Specific fixed benthic organisms have previously been used as indicators for the monitoring of various environmental parameters; such species have been used as indicators of global warming and climate change [63, 121–124], sea level fluctuations [125,126], and the influence of recreational diving on MPAs [62, 83, 127,128].

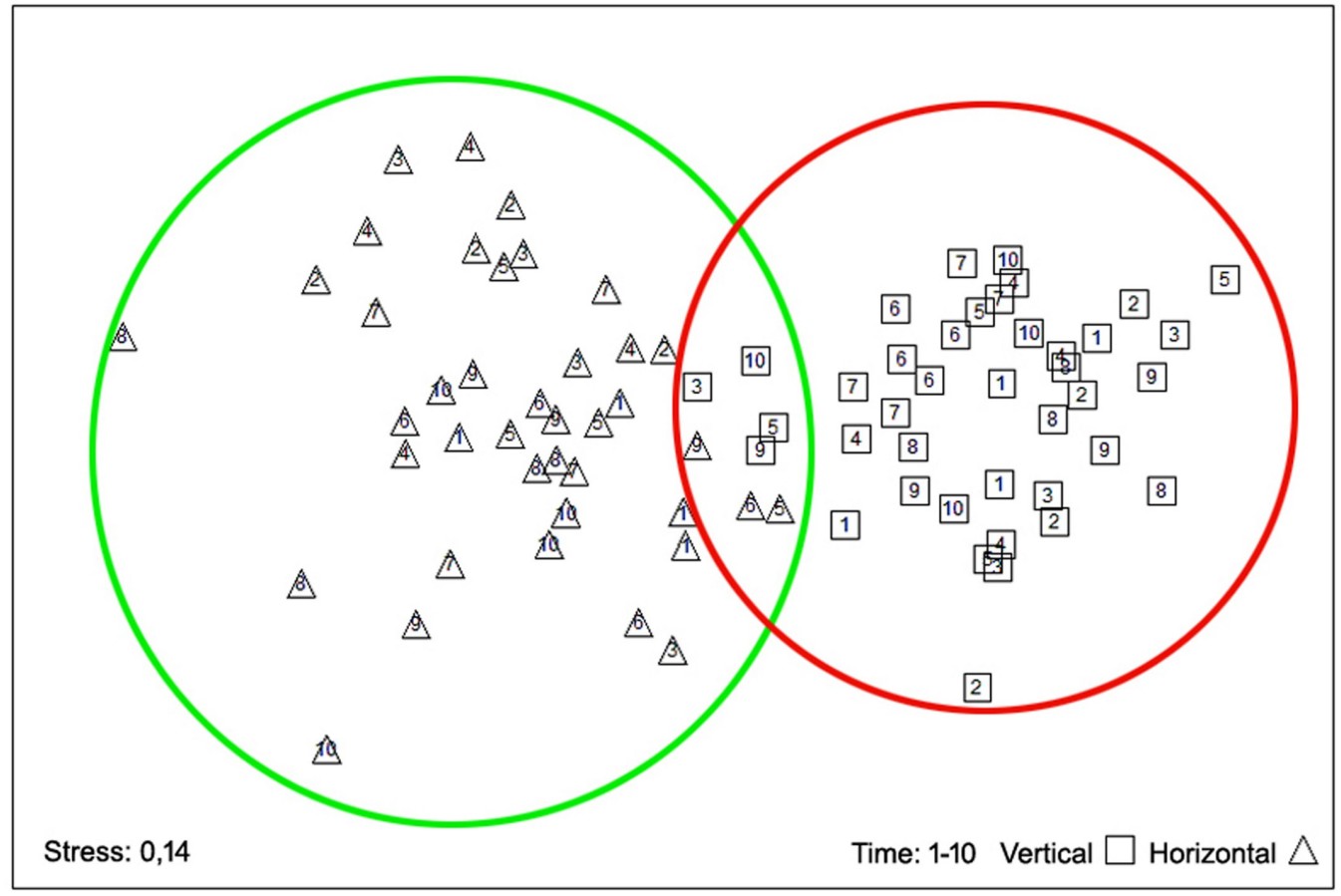

**Fig 9. Graphic representation of the non-metric MDS analysis for the whole monitoring quadrats.**

Moreover, the use of representative taxa as proxies for entire communities has proven to be a reliable alternative for evaluating the state of rocky bottom communities comprising algae because the loss of information associated with the identification of only some specific taxa does not greatly alter the results compared with results obtained with complete datasets based on entire communities [3]. In hard bottom invertebrate communities, it has also been found that sampling efforts can be reduced without significant losses of information via the selection of indicator and representative species [129].

*A. calycularis*, *P. clavata*, and *Eunicella* sp. (which were clearly more abundant in the monitored vertical enclaves) play an important structural role in coralligenous assemblages. These species colonise both horizontal bottoms and vertical walls and offer ideal habitats for numerous other organisms [43, 130, 131]. Several previous studies of coralligenous zones in different areas of the Mediterranean, such as the coasts of Italy and France, have determined that the facies of *Paramuricea* and *Eunicella* are distinctive [87, 132].

None of these species exhibited significant cover alterations over the entire monitoring period. Although *P. clavata* and *Eunicella sp.* have similar biological and ecological characteristics, including similar rates of growth and production [133–135], this similarity does not extend to their turnover rates. For *P. clavata*, this rate oscillates between 7 and 9 years [133,135], whereas for *Eunicella* sp., the rate is significantly lower with a range of 3 to 4 years [134]. For this reason, during the long-term temporal monitoring of fixed, limited points, we would expect *Eunicella* sp. to exhibit greater oscillations in cover percentages than those that we observed due to mortality among the colonies within the monitoring quadrats. Nevertheless, this phenomenon was not recorded in our work, possibly because the magnitude of the studied area allowed for the replacement of the dead organisms by new recruits to buffer the potential differences.

Also, it is important to point out that the two main indicator species, *A. calycularis* and *P. clavata* (particularly the latter), are sensitive to changes in normal temperature conditions [122, 136–138], although they both withstand seasonal fluctuations well [120].

The differences found in the covers of certain species can also be explained by natural processes. *Mesophyllum alternans* is among the organisms responsible for the greatest percentage of cover (particularly in the horizontal orientation), but it is considered to be opportunistic [51], which might explain any oscillations in cover in the absence of any significant alterations to the reference species. The other two species that exhibited significant differences, *i.e.*, *Salmacina incrustans* and *Crambe* sp., are organisms that account for very low percentages of the overall cover. Variations in abundances in deep rocky bottom communities are usually much more noticeable among minor species than among the highly abundant or characteristic species [139]. In any case, these two species can be classified as tolerant and relatively resistant to environmental impacts [43].

Recent studies of coralligenous areas have only used vertical enclaves [132], but, despite comprising the same species, the biological communities that occur at the two orientations are clearly different due to the varying percentages of cover formed by those species. Encrusting algae dominate horizontal enclaves, whereas populations of *P. clavata* and *A. calycularis* are more abundant on vertical walls; these differences have previously been mentioned by other authors [120, 140]. Therefore, wherever possible, stations should be set up in both orientations, although vertical coralligenous enclaves provide more information because they have greater biodiversity, are better structured spatially and trophically, and have greater numbers of sensitive colonial species.

## Applications of fixed quadrats for the monitoring of rocky bottom habitats

The method tested here would be particularly useful for the WFD/MSFD in relation to the environmental control of littoral zones based on biological indicators. The WFD/MSFD has

very few methodological tools for monitoring sessile benthic invertebrates associated with hard substrates, although the tools that do exist are excellent [141]. The environmental information provided by these sessile benthic species is significant because it covers a wide range of sessile organisms that are long-lived and sensitive to changes in the environment. These species can be used as reliable ecological sentries that keep watch over the quality of littoral environments because they are not able to flee or be displaced (as adults) when conditions deteriorate or significantly change.

The effectiveness of this monitoring method was demonstrated over the 10-year period used to obtain a complete quantitative temporal series. As indicated by Gatti [87,142], such series are very useful for evaluating both reference conditions and environmental changes in ecosystems. Indeed, the main macrobiota species in the sublittoral community in this study were stable and remained in excellent condition from 2005 to 2014.

The use of images is advantageous because, unlike the removal of physical samples by scraping, it is neither destructive nor aggressive to the surroundings. The scraping approach is not effective as a monitoring method in zones in which the fauna is distributed in patches because it entails the destruction of relatively large areas [106]. In protected locations and areas of special interest, this scraping option is particularly undesirable.

The present image-based monitoring method is "low cost" in addition to being "low effort" and requires minimal maintenance (repairs or replacements of the fixed quadrats were necessary only twice over the 10-year monitoring period). Panayotidis [22] indicated that the use of simple monitoring programs (*i.e.*, those involving taxonomic efforts that are limited to representative species and simple statistical treatments) and low budget methods is demonstrably effective for other rocky bottom communities.

According to recent studies, the minimum replication area required for biodiversity studies of communities dominated by *Paramuricea clavata* [143], as well as other hard-substrate benthic invertebrates [92, 144,145], is less than the area used in the replicates in our present work. Additionally, the captured images remain available for subsequent, more detailed descriptions of the community. Higher quality images facilitate the visual identification of more of the macrobenthic species that are present in the community, and the system of photo-quadrats is more efficient than *in situ* visual census methods [92] while simultaneously avoiding the destruction of the monitored areas.

As a final summary, Table 3 presents a comparison of the data from the monitoring area of the present study with the data from several other studies of coralligenous bottoms in terms of time and depth. Accounting for both area and time, the present study represents a long-term monitoring of coralligenous assemblages.

**Table 3. Comparison of the data of monitoring area, time and depth of several studies on coralligenous bottoms.**

| Study | | Location | Area (m$^2$) | Depth (m) | Time (years) |
|---|---|---|---|---|---|
| [139] | Peckol P (1984) | North Carolina | 0.132 | 20 | 2 |
| [51] | Garrabou J (2000) | Medes Islands | 1.085 | 15–19 | 2 |
| [44] | Garrabou J (2002) | Marseilles, France | 0.4 | 27 | 21 |
| [66] | Garrabou J (2002) | NW Mediterranean | 0.372 | 17–20 | 2 |
| [146] | De Biasi A M (2004) | Aegean Sea | 33.6 | 10–35 | 1 |
| [112] | Bussotti S (2006) | Southern Italy | 13.25 | 6–8 | 1 |
| [147] | Parravicini V (2009) | Ligurian Sea | 25 | 4–5 | 1 |
| [36] | Piazzi L (2017) | Ligurian Sea | 180 | 30–40 | 1 |
| [92] | Sant N (2017) | Cabrera Archipelago | 4.96 | 0–50 | 1 |
| Present study | | | 8 | 27–30 | 10 |

Finally, the proposed methodology could be extended to the entire geographical area over which the target species of this study are representative of the rocky bottom benthic communities that occur at similar depths and share similar biotic structures. This methodology could also be applied in other areas in which target species can be established and selected by employing the aforementioned implementation criteria. Additionally, the SBPQ method involves minimal implementation difficulty both in terms of cost and installation. The *a priori* selection of the target species simplifies the taxonomic difficulty of visual identification. Therefore, this environmental warning and surveillance system is not only oriented toward the scientific and technical communities of the relevant coastal countries but also toward environmental volunteers associated with diving centres and clubs. This methodology ultimately entails the aim of setting up geographical networks in regions with sufficiently high levels of homogeneity (*e.g.*, the Alboran Sea or, on a larger scale, the western Mediterranean). Citizen science initiatives focusing on the mapping and monitoring of coralligenous assemblages have recently been implemented [148]. In the future, these networks could provide early warnings of changes to structured and unpolluted systems, particularly those in littoral zones close to places subjected to strong anthropogenic pressure, which could be applied to the detection of local or general environmental alterations (*i.e.*, climate change). In contrast, the nature of the method, which is designed to monitor a group of taxa with respect to the total taxa of the community, implies that it is not useful as such in the ecological study of these habitats because the analysed information is biased. However, the generated databases (*i.e.*, stored photographs) will allow for the performance of in-depth studies of the dynamics of the species within the communities over time, assuming that significant changes can be detected, and could aid parallel projects that aim to increase our knowledge of coralligenous assemblages.

## Conclusions

The study assessed the Sessile Bioindicators in Permanent Quadrats (SBPQ) underwater environmental alert method in relation to the monitoring of pre-selected sensitive and sessile benthic species (*P. clavata* and *A. calycularis*) associated with rocky coralligenous habitats (which exhibit high stability and biodiversity) via the use of permanent quadrats fixed on rocky shores in underwater SBPQ sentinel stations. The obtained results have allowed for assessing the part of the method based on the initial hypothesis that, over a long temporal duration (a ten-year period in this study) and in a highly structured and biodiverse coralligenous assemblage, the cover of sensitive sessile species does not change over time if the environmental stability characterising the habitat is not altered. This stability of the cover of the sensitive sessile species is a key aspect for confirming the reliability and robustness of the SBPQ method. Given that, as in the present study, the selected species are very sensitive to increases in temperature and deterioration of the environmental quality of the water column, the SBQP method is useful as an underwater environmental alert system because it should be solely sensitive to changes in the coverages of such species that result from physico-chemical changes in the system. Such changes include gradual increases in temperature due to global warming and changes due to the introduction of exotic species.

A future usefulness would be to implement the SBPQ methodology in well-conserved areas so those areas can act as "SBPQ sentinel stations" in the event of possible disturbances. The method has been developed as a simple management tool for use by scientists and specialised technicians in addition to diving clubs that frequent certain areas.

## Supporting information

**S1 Table. Coverage of each species through the monitoring period.** Time: 1 corresponding to 2005 and 10 to 2014. Slope: H, horizontal; V, vertical.
(XLS)

## Acknowledgments

We want to warmly thank the Campo de Gibraltar, Centro de Investigaciones y Exploraciones Submarinas CIES Algeciras and CIES Tarifa Diving Centers, for their joint participation with the authors in a general "Citizen Science" program related to underwater sentinel stations. Also, we express our gratitude for the collaboration provided to the Seville Aquarium, Town Halls of Tarifa and La Línea as well as to Puerto Deportivo La Alcaidesa, Club Marítimo y Real Club Náutico (RCN) de La Línea. Finally, we would like to mention Academic Editor PLOS ONE and the reviewers of this paper for their valuable recommendations, which improved the contents of this work.

## Author Contributions

**Conceptualization:** José C. García-Gómez.

**Data curation:** Alexandre R. González.

**Formal analysis:** José C. García-Gómez, Alexandre R. González.

**Funding acquisition:** José C. García-Gómez.

**Investigation:** José C. García-Gómez, Alexandre R. González, Manuel J. Maestre, Free Espinosa.

**Methodology:** José C. García-Gómez.

**Project administration:** José C. García-Gómez.

**Resources:** José C. García-Gómez.

**Software:** Alexandre R. González.

**Supervision:** José C. García-Gómez.

**Validation:** José C. García-Gómez, Alexandre R. González, Manuel J. Maestre, Free Espinosa.

**Visualization:** José C. García-Gómez, Alexandre R. González.

**Writing – original draft:** José C. García-Gómez, Alexandre R. González.

**Writing – review & editing:** José C. García-Gómez, Alexandre R. González, Manuel J. Maestre, Free Espinosa.

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
