## [Decision Letter · Decision Letter 0]

28 Oct 2019

PONE-D-19-25987

Long-term pattern of sessile rocky bottom bioindicators. Testing early-warning system for detecting coastal disturbances and climate change effects

PLOS ONE

Dear Dr. Maestre,

Thank you for submitting your manuscript to PLOS ONE. After careful consideration, we feel that it has merit but does not fully meet PLOS ONE’s publication criteria as it currently stands. Therefore, we invite you to submit a revised version of the manuscript that addresses the points raised during the review process.

Both reviewers suggested additional references and asked for further methodological detail.

We would appreciate receiving your revised manuscript by Dec 12 2019 11:59PM. To enhance the reproducibility of your results, we recommend that if applicable you deposit your laboratory protocols in protocols.io, where a protocol can be assigned its own identifier (DOI) such that it can be cited independently in the future. For instructions see: http://journals.plos.org/plosone/s/submission-guidelines#loc-laboratory-protocols

We look forward to receiving your revised manuscript.

Kind regards,

Carlo Nike Bianchi

Academic Editor

PLOS ONE

Journal Requirements:

Reviewers' comments:

Reviewer's Responses to Questions

**Comments to the Author**

1. Is the manuscript technically sound, and do the data support the conclusions?

Reviewer #1: Partly

Reviewer #2: Partly

2. Has the statistical analysis been performed appropriately and rigorously? 

Reviewer #1: Yes

Reviewer #2: I Don't Know

3. Have the authors made all data underlying the findings in their manuscript fully available?

Reviewer #1: Yes

Reviewer #2: No

4. Is the manuscript presented in an intelligible fashion and written in standard English?

Reviewer #1: Yes

Reviewer #2: Yes

5. Review Comments to the Author

Reviewer #1: I think that the work and the data presented by the authos are very important. One decade of data on coralligenous is rare in Mediterranean. However, all this great work is not supported by the manuscript.

I had some pluridecadal experience on the subject and I hope that the following suggestions could help to rewrite the manuscript.

1) the RAC-SPa manual cited (41) is a great work but the method it should be summarized and described for non-mediterranean scientist. I think it should be also of great importance to briefly introduce also the RAC-SPA and its work for non-Mediterranean scientist.

2) the photographic method has been applied since at least 30 years from the use of the first

underwater photographic apparatus and the ''sensitive' species has been described since 1958 by Peres and Picard in their ' Manuel de bionomie bentonique' etc. etc..

I think that the introduction should be rewritten considering also that a lot of scientitst in these last years, have worked on the coralligenous and to the creation of indices. You cited many papers on the subject this but do not describe the differences between their approach and your.

I found also these papers:

-Julie Deter a,b,⁎, Pierre Descamp c, Pierre Boissery d, Laurent Ballesta c, Florian Holon cA rapid photographic method detects depth gradient in coralligenous assemblages.

Journal of Experimental Marine Biology and Ecology 418–419 (2012) 75–82

-Silvija Kipson1,3*, Maı¨a Fourt2, Nu´ ria Teixido´ 1,6, Emma Cebrian4, Edgar Casas1, Enric Ballesteros5, Mikel

Zabala6, Joaquim Garrabou. Rapid Biodiversity Assessment and Monitoring Method for Highly Diverse Benthic Communities: A Case Study of Mediterranean Coralligenous Outcrops PLoS ONE | www.plosone.org 1 November 2011 | Volume 6 | Issue 11 | e27103

-Giulia Gattia*, Monica Montefalconea, Alessio Rovereb, Valeriano Parravicinibc, Carla Morria, Giancarlo Albertellia and Carlo Nike Bianchi.Seafloor integrity down the harbor waterfront: the coralligenous shoals off Vado Ligure (NW Mediterranean)Advances in Oceanography and Limnology Vol. 3, No. 1, June 2012, 51–67

-Luigi Piazzi1 | Carlo Nike Bianchi2 | Enrico Cecchi3 | Giulia Gatti4 | Ivan Guala5 |

Carla Morri2 | Stéphane Sartoretto6 | Fabrizio Serena3,7 | Monica Montefalcone2 What's in an index? Comparing the ecological information provided by two indices to assess the status of coralligenous reefs in the NW Mediterranean SeaAquatic Conserv: Mar Freshw Ecosyst. 2017;27:1091–1100

Hence, I suggest to the author to have the patience to modify the introduction, the material and methods and use the available published papers on the subject to compare their method and to evaluate the pros and cons.

Andrea Peirano

Reviewer #2: Dear Editor, please find attached my comments on the paper “Long-term pattern of sessile rocky bottom bioindicators. Testing early-warning system for detecting coastal disturbances and climate change effects” by García-Gómez et al.

Developing methods to detect environmental alterations is currently a hot topic, especially in Europe within the MSFD. The present study aims to validate a new methodology by verifying that long-term cover of indicator species does not change in undisturbed ecosystems. In my opinion, this manuscript presents some shortcomings which prevent its publication in its current status. Some explanations follow.

a) I understand that the SBPQ method here proposed is simple, low-cost and easy to use, and I support the idea of a network of sentry stations monitored with fixed quadrats. Anyway, as already mentioned by the authors (line 420), other studies have recently provided excellent tools to monitor benthic ecosystems. Thus, a more robust justification to develop a new one should be included.

b) Authors state that “the SBPQ method was proposed by García-Gómez [41], but its validation is pending a study of a long time series that could confirm that the coverages of long-cycle indicator species do not change over time in undisturbed benthic environments with high levels of biodiversity unless a significant environmental alteration is introduced into and modifies the system” (lines 162-166). I am not convinced that it is enough for the complete validation of the method. This study could prove that the basic idea of monitoring the cover of sensitive indicator species through time is useful, but not that the method itself is able to properly detect environmental changes.

No other aspect of this method has been previously validated somewhere else through a peer review process. I suggest to include a more detailed description of the SBPQ methodology within the M&M section, providing not only information on target species and sampling procedure but also on the outputs processing, which are currently missing. I can see that in García-Gómez [41] a scale has been set out to detect significative environmental changes (<25%, no impact; 25-50%, orange warning sign; >50%, red warning sign; page 115). This is an interesting point and it has to be explained how these values have been selected. In my opinion, a loss of 25% cover in a species that cover on average 30-35% of the quadrant (like P. clavata does in the present study, Fig. 4) is more than considerable.

The best way to validate this method is to compare different cases characterized by different ecological settings. This would be especially useful to set the scale proposed in García-Gómez [41]. Anyway, I understand that it is really complicated to perform it now, so I wonder if it is possible to collect sufficient information from literature on sensitive species cover changing in relation to environmental gradients. The authors of this paper referred to several studies dealing with this problem and some additional references are provided below.

c) The authors sustain that environmental stability of the investigated habitat is not changed during this study, and this represents a key point for the method validation, since sensitive species cover is expected to decrease only if i) coastal disturbances and ii) climate change occur. Temperature time series have been reported from the study area, confirming the stability of the environment, but no information has been reported regarding nutrients or pollutant concentration. This information is essential because the study area lye proximal to zones under strong anthropic pressure (es. Bay of Algeciras) and it is subjected to a high level of marine traffic and polluting events.

Hence, without considering all these shortcomings, in my opinion, this paper cannot be published.

Other comments:

Line 65. I think this reference could be of interest:

de Juan, S., & Demestre, M. (2012). A Trawl Disturbance Indicator to quantify large scale fishing impact on benthic ecosystems. Ecological Indicators, 18, 183-190.

Line 71. Please consider this reference:

Gobert, S., Sartoretto, S., Rico-Raimondino, V., Andral, B., Chery, A., Lejeune, P., & Boissery, P. (2009). Assessment of the ecological status of Mediterranean French coastal waters as required by the Water Framework Directive using the Posidonia oceanica Rapid Easy Index: PREI. Marine Pollution Bulletin, 58(11), 1727-1733.

Line 82-83. I agree that short-term studies are more common than long-term ones, but nowadays long-term studies are becoming not so rare. Here I report some additional examples.

Bianchi, C. N., Cocito, S., Diviacco, G., Dondi, N., Fratangeli, F., Montefalcone, M., ... & Morri, C. (2018). The park never born: Outcome of a quarter of a century of inaction on the sea‐floor integrity of a proposed but not established Marine Protected Area. Aquatic Conservation: Marine and Freshwater Ecosystems, 28(5), 1209-1228.

Betti, F., Bavestrello, G., Bo, M., Asnaghi, V., Chiantore, M., Bava, S., & Cattaneo‐Vietti, R. (2017). Over 10 years of variation in Mediterranean reef benthic communities. Marine Ecology, 38(3), e12439.

[130] Montefalcone, M., Morri, C., Bianchi, C. N., Bavestrello, G., & Piazzi, L. (2017). The two facets of species sensitivity: Stress and disturbance on coralligenous assemblages in space and time. Marine pollution bulletin, 117(1-2), 229-238.

Bertolino, M., Betti, F., Bo, M., Cattaneo-Vietti, R., Pansini, M., Romero, J., & Bavestrello, G. (2016). Changes and stability of a Mediterranean hard bottom benthic community over 25 years. Journal of the Marine Biological Association of the United Kingdom, 96(2), 341-350.

Line 104. I suggest changing with “Study area”.

Line 160-181. This paragraph reports several repetitions and inaccurate information. I suggest to delete it.

Line 161-166. This information has been already provided.

Line 166-172. Copied and pasted from lines 105-112.

Line 175-176. I suggest moving this sentence in the next paragraph.

Line 176-181. I understand that a summary of the SBPQ method could help readers in understanding the ms. Anyway, I found this part confusing. Please delete or rephrase it.

Line 203. Only one location is mentioned in lines 141-142. Please clarify this part.

Line 208. Please provide more information on the month/season selected for annual sampling.

Line 267. In order to help readers, I suggest separating the results from the discussion sections.

Line 283. In my opinion, the differences observed in Fig. 4 are significant. I understand that the RM-ANOVA does not detect significative differences, but I can clearly see that A. calycularis (V) cover increases from about 10-15% to 25-30% and P.clavata (V) cover decreases from about 30-35% to 20-25%.

Please note that if the decreasing trend showed by P. clavata is maintained, this species is expected to disappear in the next 20-30 years. Furthermore, a similar decreasing trend in P. clavata coverage over 10 years has been observed by Betti et al., 2017 and I think it should be properly considered in the discussion section.

Line 314. I understand the importance to demonstrate that the orientation factor deeply influences coralligenous assemblages, but this is not the main objective of this paper. This evidence is highlighted by the RM-ANOVA and RM-PERMANOVA analyses (Tables 1 and 2) and from Figures 3 and 4. Thus, I think that the nmMDS (Fig. 5) is redundant. Have authors considered any multivariate statistic method to verify that there are no significative differences among species cover in vertical quadrats?

Line 320. A “Discussion” section could start here.

Line 419-420. Please see my previous comment (a).

Line 421. “This species” is not clear in this sentence.

Line 461. Table 3. I suggest inserting an additional column with each geographical location. I also suggest to report first author name and year of publication for the studies in the first column.

I strongly suggest to include a supplementary material table, including all the data collected during this study. Authors could use two SM tables, one for each orientation (vertical/horizontal), to report the mean values of coverage for the target species during the 10 years.

A total of 320 pictures were analyzed in this study. I suggest to include a new figure in the main text, showing a picture taken in 2005 and the same picture taken in 2014. This could be done for both vertical and horizontal quadrats, for a total of 4 pictures in the figure.

6. PLOS authors have the option to publish the peer review history of their article (what does this mean?). If published, this will include your full peer review and any attached files.

Reviewer #1: Yes: Andrea Peirano

Reviewer #2: No

---

## [Author Response · Author response to Decision Letter 0]

4 Jan 2020

Responses to reviewers

Reviewer #1: I have incorporated your suggestions into my revision. Thank for your help. Below I present your numbered comments and corresponding response.

Comment 1:

The RAC-SPa manual cited (41) is a great work but the method it should be summarized and described for non-mediterranean scientist. I think it should be also of great importance to briefly introduce also the RAC-SPA and its work for non-Mediterranean scientist.

Answer to comment number 1:

The methodology has been expanded and a more detailed description of the SBPQ method has been included. In addition, two figures have been added to illustrate it. The RACSPA has also been briefly presented.

Comment 2:

The photographic method has been applied since at least 30 years from the use of the first

underwater photographic apparatus and the ''sensitive' species has been described since 1958 by Peres and Picard in their ' Manuel de bionomie bentonique' etc. etc..

I think that the introduction should be rewritten considering also that a lot of scientitst in these last years, have worked on the coralligenous and to the creation of indices. You cited many papers on the subject this but do not describe the differences between their approach and your.

I found also these papers:

-Julie Deter a,b,⁎, Pierre Descamp c, Pierre Boissery d, Laurent Ballesta c, Florian Holon cA rapid photographic method detects depth gradient in coralligenous assemblages.

Journal of Experimental Marine Biology and Ecology 418–419 (2012) 75–82

-Silvija Kipson1,3*, Maı¨a Fourt2, Nu´ ria Teixido´ 1,6, Emma Cebrian4, Edgar Casas1, Enric Ballesteros5, Mikel

Zabala6, Joaquim Garrabou. Rapid Biodiversity Assessment and Monitoring Method for Highly Diverse Benthic Communities: A Case Study of Mediterranean Coralligenous Outcrops PLoS ONE | www.plosone.org 1 November 2011 | Volume 6 | Issue 11 | e27103

-Giulia Gattia*, Monica Montefalconea, Alessio Rovereb, Valeriano Parravicinibc, Carla Morria, Giancarlo Albertellia and Carlo Nike Bianchi.Seafloor integrity down the harbor waterfront: the coralligenous shoals off Vado Ligure (NW Mediterranean)Advances in Oceanography and Limnology Vol. 3, No. 1, June 2012, 51–67

-Luigi Piazzi1 | Carlo Nike Bianchi2 | Enrico Cecchi3 | Giulia Gatti4 | Ivan Guala5 |

Carla Morri2 | Stéphane Sartoretto6 | Fabrizio Serena3,7 | Monica Montefalcone2 What's in an index? Comparing the ecological information provided by two indices to assess the status of coralligenous reefs in the NW Mediterranean SeaAquatic Conserv: Mar Freshw Ecosyst. 2017;27:1091–1100

Answer to comment number 2:

A paragraph has been included describing the differences between the proposed methodology and that designed by other authors. The articles suggested by the reviewer have been taken into account and cited.

Reviewer #2: I have incorporated your suggestions into my revision. Thank you for your contribution. It has been very useful. Below I present your numbered comments and corresponding response.

Comment 1:

I understand that the SBPQ method here proposed is simple, low-cost and easy to use, and I support the idea of a network of sentry stations monitored with fixed quadrats. Anyway, as already mentioned by the authors (line 420), other studies have recently provided excellent tools to monitor benthic ecosystems. Thus, a more robust justification to develop a new one should be included.

Answer to comment number 1:

As previously mentioned, a justification of the proposal has been included in the introduction section.

Comment 2:

Authors state that “the SBPQ method was proposed by García-Gómez [41], but its validation is pending a study of a long time series that could confirm that the coverages of long-cycle indicator species do not change over time in undisturbed benthic environments with high levels of biodiversity unless a significant environmental alteration is introduced into and modifies the system” (lines 162-166). I am not convinced that it is enough for the complete validation of the method. This study could prove that the basic idea of monitoring the cover of sensitive indicator species through time is useful, but not that the method itself is able to properly detect environmental changes.

Answer to comment number 2:

The citation of a recently published article has been included. This study shows the validity of the method to detect disturbances for invasive species.

García-Gómez J, Sempere-Valverde J, González A, Martínez-Chacón M, Olaya-Ponzone L, Sánchez-Moyano E, Ostalé Valriberas E, Megina C. From exotic to invasive in record time: the extreme impact of Rugulopteryx okamurae (dictyotales, Ochrophyta) in the strait of Gibraltar. Science of The Total Environment. 704 (2020) 135408.

Comment 3:

No other aspect of this method has been previously validated somewhere else through a peer review process. I suggest to include a more detailed description of the SBPQ methodology within the M&M section, providing not only information on target species and sampling procedure but also on the outputs processing, which are currently missing. I can see that in García-Gómez [41] a scale has been set out to detect significative environmental changes (<25%, no impact; 25-50%, orange warning sign; >50%, red warning sign; page 115). This is an interesting point and it has to be explained how these values have been selected. In my opinion, a loss of 25% cover in a species that cover on average 30-35% of the quadrant (like P. clavata does in the present study, Fig. 4) is more than considerable.

Answer to comment number 3:

The method has been described in detail and the references used to establish the defined coverage percentage classes have been included. In addition, the methodology was partially validated in the previously mentioned study (García-Gómez et al., 2020).

Comment 4:

The best way to validate this method is to compare different cases characterized by different ecological settings. This would be especially useful to set the scale proposed in García-Gómez [41]. Anyway, I understand that it is really complicated to perform it now, so I wonder if it is possible to collect sufficient information from literature on sensitive species cover changing in relation to environmental gradients. The authors of this paper referred to several studies dealing with this problem and some additional references are provided below.

Answer to comment number 4:

The proposed method could be easily used at different geographical scales or under different ecological settings identifying the appropriate target species in the way describe on lines 543-547. In fact, one of the proposed target species for our study area (the red gorgonia Paramuricea clavata) showed sensitivity for thermal anomalies in different places such as Balearic Islands (Linares et al., 2017). Nevertheless, it is important to disentangle sensitivity itself from other potential confounding factors such as population dynamics showing sharp changes in coverage through the time not related to environmental disturbances. It is the main objective of the study, i.e., demonstrate that the selected species remain stable in coverage in a long-time period.

Linares C, Ballesteros E, Verdura J, Aspillaga E, Capdevila P, Coma R et al. Efectos del cambio climático sobre la gorgonia Paramuricea clavata y el coralígeno asociado en el parque nacional marítimo-terrestre del archipiélago de cabrera. Proyectos de investigación en Parques Nacionales: 2012-2015.

Comment 5:

The authors sustain that environmental stability of the investigated habitat is not changed during this study, and this represents a key point for the method validation, since sensitive species cover is expected to decrease only if i) coastal disturbances and ii) climate change occur. Temperature time series have been reported from the study area, confirming the stability of the environment, but no information has been reported regarding nutrients or pollutant concentration. This information is essential because the study area lye proximal to zones under anthropic pressure (es. Bay of Algeciras) and it is subjected to a high level of marine traffic and polluting events.

Answer to comment number 5:

The species were selected for their sensitivity to chemical contamination and other sources of environmental perturbation based on other previous studies that have been properly referenced. Therefore, the relationship between absence of industrial or sewage pollution and the presence of these target species is a well established evidence in the scientific literature. As we have indicated, the objective of the study was to determine that the coverage of these species remain stable in the long term under normal environmental conditions. No significant contamination event has been reported for this period in the study area according to the available information by environmental authorities.

Comment 6:

Line 65. I think this reference could be of interest:

de Juan, S., & Demestre, M. (2012). A Trawl Disturbance Indicator to quantify large scale fishing impact on benthic ecosystems. Ecological Indicators, 18, 183-190.

Line 71. Please consider this reference:

Gobert, S., Sartoretto, S., Rico-Raimondino, V., Andral, B., Chery, A., Lejeune, P., & Boissery, P. (2009). Assessment of the ecological status of Mediterranean French coastal waters as required by the Water Framework Directive using the Posidonia oceanica Rapid Easy Index: PREI. Marine Pollution Bulletin, 58(11), 1727-1733.

Line 82-83. I agree that short-term studies are more common than long-term ones, but nowadays long-term studies are becoming not so rare. Here I report some additional examples.

Bianchi, C. N., Cocito, S., Diviacco, G., Dondi, N., Fratangeli, F., Montefalcone, M., ... & Morri, C. (2018). The park never born: Outcome of a quarter of a century of inaction on the sea‐floor integrity of a proposed but not established Marine Protected Area. Aquatic Conservation: Marine and Freshwater Ecosystems, 28(5), 1209-1228.

Betti, F., Bavestrello, G., Bo, M., Asnaghi, V., Chiantore, M., Bava, S., & Cattaneo‐Vietti, R. (2017). Over 10 years of variation in Mediterranean reef benthic communities. Marine Ecology, 38(3), e12439.

[130] Montefalcone, M., Morri, C., Bianchi, C. N., Bavestrello, G., & Piazzi, L. (2017). The two facets of species sensitivity: Stress and disturbance on coralligenous assemblages in space and time. Marine pollution bulletin, 117(1-2), 229-238.

Bertolino, M., Betti, F., Bo, M., Cattaneo-Vietti, R., Pansini, M., Romero, J., & Bavestrello, G. (2016). Changes and stability of a Mediterranean hard bottom benthic community over 25 years. Journal of the Marine Biological Association of the United Kingdom, 96(2), 341-350.

Line 104. I suggest changing with “Study area”.

Line 160-181. This paragraph reports several repetitions and inaccurate information. I suggest to delete it.

Line 161-166. This information has been already provided.

Line 166-172. Copied and pasted from lines 105-112.

Line 175-176. I suggest moving this sentence in the next paragraph.

Line 176-181. I understand that a summary of the SBPQ method could help readers in understanding the ms. Anyway, I found this part confusing. Please delete or rephrase it.

Line 203. Only one location is mentioned in lines 141-142. Please clarify this part.

Answer to comment number 6:

All suggestions have been taken into account in the revised manuscript. The proposed references have been included.

Comment 7:

Line 208. Please provide more information on the month/season selected for annual sampling.

Answer to comment number 7:

All samples were performed between the spring and summer seasons.

Comment 8:

Line 267. In order to help readers, I suggest separating the results from the discussion sections.

Answer to comment number 8:

It has been done.

Comment 9:

Line 283. In my opinion, the differences observed in Fig. 4 are significant. I understand that the RM-ANOVA does not detect significative differences, but I can clearly see that A. calycularis (V) cover increases from about 10-15% to 25-30% and P.clavata (V) cover decreases from about 30-35% to 20-25%.

Please note that if the decreasing trend showed by P. clavata is maintained, this species is expected to disappear in the next 20-30 years. Furthermore, a similar decreasing trend in P. clavata coverage over 10 years has been observed by Betti et al., 2017 and I think it should be properly considered in the discussion section.

Answer to comment number 9:

The following sentence has been included in the discussion section (line 356). "The result of the analysis shows a decrease in the coverage of P. clavata throughout the studied period. This circumstance coincides with that observed in other studies (Betty et al., 2017). Environmental factors such as climate change may be affecting certain species more slowly. These trends may be analyzed when they have a longer period.”

Comment 10:

Line 314. I understand the importance to demonstrate that the orientation factor deeply influences coralligenous assemblages, but this is not the main objective of this paper. This evidence is highlighted by the RM-ANOVA and RM-PERMANOVA analyses (Tables 1 and 2) and from Figures 3 and 4. Thus, I think that the nmMDS (Fig. 5) is redundant. Have authors considered any multivariate statistic method to verify that there are no significative differences among species cover in vertical quadrats?

Answer to comment number 10:

The two multivariate statistic methods used has been RM-PERMANOVA and nmMDS. The latter, unlike the first, is an indirect gradient analysis approach which produces an ordination based on a distance or dissimilarity matrix and, although it may be expendable as the reviewer suggests, we recommend its inclusion in the paper as it is a very visual graph, and serves as a reinforcement of the RM-ANOVA analytics and RM-PERMANOVA carried out. However, we are opened to include other different multivariate analysis that the referee considers appropriate to suggest.

Comment 11:

Line 320. A “Discussion” section could start here.

Answer to comment number 11:

It has been done.

Comment 12:

Line 419-420. Please see my previous comment (a).

Answer to comment number 12:

Our methodology is low cost and its implementation is simple. The ultimate goal is to create a network in which not only scientists and technicians participate, but also a diving club or volunteer groups.

Comment 13:

Line 421. “This species” is not clear in this sentence.

Answer to comment number 13:

The environmental information provided by these sessile benthic species is significant because it covers a wide range of sessile organisms that are long-lived and sensitive to changes in the environment.

Comment 14:

Line 461. Table 3. I suggest inserting an additional column with each geographical location. I also suggest to report first author name and year of publication for the studies in the first column.

Answer to comment number 14:

The table 3 has been inserted an additional column with each geographical location and first author name and year of publication for the studies has been reported in the first column.

Comment 15:

I strongly suggest to include a supplementary material table, including all the data collected during this study. Authors could use two SM tables, one for each orientation (vertical/horizontal), to report the mean values of coverage for the target species during the 10 years.

Answer to comment number 15:

A supplementary material table has been included.

Comment 16:

A total of 320 pictures were analyzed in this study. I suggest to include a new figure in the main text, showing a picture taken in 2005 and the same picture taken in 2014. This could be done for both vertical and horizontal quadrats, for a total of 4 pictures in the figure.

Answer to comment number 16:

 A new figure (figure 7) that showing the pictures requested.

---

## [Decision Letter · Decision Letter 1]

10 Feb 2020

PONE-D-19-25987R1

Detect coastal disturbances and climate change effects in coralligenous community through  sentinel stations

PLOS ONE

Dear Dr. Maestre,

Thank you for submitting your manuscript to PLOS ONE. After careful consideration, we feel that it has merit but does not fully meet PLOS ONE’s publication criteria as it currently stands. Therefore, we invite you to submit a revised version of the manuscript that addresses the points raised during the review process.

The reviewer laments that their critics have been not taken in full account. Please consider the possibility that your revised version and accompanying letter are sent again to the reviewer.

We would appreciate receiving your revised manuscript by Mar 26 2020 11:59PM. To enhance the reproducibility of your results, we recommend that if applicable you deposit your laboratory protocols in protocols.io, where a protocol can be assigned its own identifier (DOI) such that it can be cited independently in the future. For instructions see: http://journals.plos.org/plosone/s/submission-guidelines#loc-laboratory-protocols

We look forward to receiving your revised manuscript.

Kind regards,

Carlo Nike Bianchi

Academic Editor

PLOS ONE

Reviewers' comments:

Reviewer's Responses to Questions

**Comments to the Author**

1. If the authors have adequately addressed your comments raised in a previous round of review and you feel that this manuscript is now acceptable for publication, you may indicate that here to bypass the “Comments to the Author” section, enter your conflict of interest statement in the “Confidential to Editor” section, and submit your "Accept" recommendation.

Reviewer #2: (No Response)

2. Is the manuscript technically sound, and do the data support the conclusions?

Reviewer #2: Partly

3. Has the statistical analysis been performed appropriately and rigorously? 

Reviewer #2: I Don't Know

4. Have the authors made all data underlying the findings in their manuscript fully available?

Reviewer #2: Yes

5. Is the manuscript presented in an intelligible fashion and written in standard English?

Reviewer #2: Yes

6. Review Comments to the Author

Reviewer #2: Dear Editor,

I confirm that the manuscript topic is relevant and the long-term dataset of major interest. Anyway, the main shortcomings I highlighted in the previous version of this ms, regarding the methodology herein described, have not been fulfilled.

The validation of the method represents the key aspect of this work: authors state that the main goal of this paper is “to validate the Sessile Bioindicators in Permanent Quadrats (SBPQ) underwater environmental alert method” (lines 24, 120, 523). The validation of a method must be done by using totally independent dataset in which you apply the method and look at the results. Otherwise, you are using a circular argument.

The reference proposed by the authors (see the answer to my comment 2 in the previous revision) may prove that the method is able to detect the presence of invasive species, but it still does not prove that the SBPQ properly detects the several environmental changes that authors list in the text (line 174): “This methodology is a simple, non-invasive, underwater environmental alert tool for the early detection of environmental impacts of anthropic origin in the sublittoral system: in the short term (local alterations derived from pollutants from industries or emissions of urban origin, coastal dredging or civil engineering works on the coast, intrusion of exotic species with invasive potential, among other sources of alteration of coastal waters), and in the medium or long term (global warming)”.

In my opinion, the text should focus on the long-term monitoring, the 10-years dataset and the method employed to obtain it, avoiding the use of “validation” or “to confirm the untested part of the method” (line 34, 126, 528). Please note also that the description of the method itself is long and difficult to read, reporting information of scarce relevance, whereas the description of the outputs processing (which is, in my opinion, of major relevance) is still lacking (see my comment 3 in the previous revision).

7. PLOS authors have the option to publish the peer review history of their article (what does this mean?). If published, this will include your full peer review and any attached files.

Reviewer #2: No

---

## [Author Response · Author response to Decision Letter 1]

13 Mar 2020

Responses to reviewers

Reviewer #2: we have incorporated your suggestions into revisión. We want to thank you for your contributions to improve the manuscript. We present your numbered comments and corresponding response.

Comment 1:

The validation of the method represents the key aspect of this work: authors state that the main goal of this paper is “to validate the Sessile Bioindicators in Permanent Quadrats (SBPQ) underwater environmental alert method” (lines 24, 120, 523). The validation of a method must be done by using totally independent dataset in which you apply the method and look at the results. Otherwise, you are using a circular argument.

Answer to comment number 1:

The study aims to confirm the degree of coverage stability for the selected species throughout the monitored period, this is essential for the method to be effective but does not mean that it has been fully validated. It is true that it is not correct to use the term "validation" in the indicated lines. The term "validation" has been changed to "assessment" throughout the manuscript.

On the other hand, indicate that the sentry stations that have detected the presence of the Rugulopterix okamurae algae are not in the same location as those used in the present study.

Comment 2: 

The reference proposed by the authors (see the answer to my comment 2 in the previous revision) may prove that the method is able to detect the presence of invasive species, but it still does not prove that the SBPQ properly detects the several environmental changes that authors list in the text (line 174): “This methodology is a simple, non-invasive, underwater environmental alert tool for the early detection of environmental impacts of anthropic origin in the sublittoral system: in the short term (local alterations derived from pollutants from industries or emissions of urban origin, coastal dredging or civil engineering works on the coast, intrusion of exotic species with invasive potential, among other sources of alteration of coastal waters), and in the medium or long term (global warming)”.

Answer to comment number 2:

We agree with your comment. The cited reference can only demonstrate that the methodology is effective to detect invasive species but not with respect to other situations. Global warming has been included in the discussion of the manuscript because in the article commented above, it is hypothesized that this type of invasive species may be favored by climate change. Textually: a) in the text of the referred article (pag 7) : "The explosive growth of R. okamurae documented in a very short time in Ceuta also coincided with the maximum SST in the Strait of Gibraltar and nearby areas in the period between 2000 and 2017, peaking at 23.9°C in July 2015"; b) in the Abstract (pag 1): "This bloom could have been associated with the temperature peak in July 2015 and was thus possibly linked to global warming".

It is necessary to test the methodology regarding different environmental disturbances to assess the response of the method. Line 174 has been modified to be less categorical.

“This methodology has been designed as a simple, non-invasive, underwater environmental alert tool for the potential early detection of environmental impacts of anthropic origin in the sublittoral system: in the short term (local alterations derived from pollutants from industries or emissions of urban origin, coastal dredging or civil engineering works on the coast, intrusion of exotic species with invasive potential, among other sources of alteration of coastal waters), and in the medium or long term (global warming). Even though the method is able to detect the presence of invasive species, further studies are required to test the reliability of the method for detecting other potential impacts of anthropic origin”

Comment 3:

In my opinion, the text should focus on the long-term monitoring, the 10-years dataset and the method employed to obtain it, avoiding the use of “validation” or “to confirm the untested part of the method” (line 34, 126, 528).

Please note also that the description of the method itself is long and difficult to read, reporting information of scarce relevance, whereas the description of the outputs processing (which is, in my opinion, of major relevance) is still lacking (see my comment 3 in the previous revision).

Answer to comment number 3:

The use of “validation” or “to confirm the untested part of the method” has been removed in the lines indicated.

As a response to the request indicated in comment 3 in the previous revision, the method was described in detail in the previous revised manuscript. We have tried to describe all the important aspects about the design and application of the method. The references used to establish the defined coverage percentage classes also was included. We think it can be interesting that the method is described in an open access scientific journal where the proposal has been validated through a peer review process and the methodology can be easily consulted by the stakeholders. We appreciate the suggest to include a more detailed description of the SBPQ methodology within the M&M section.

---

## [Decision Letter · Decision Letter 2]

30 Mar 2020

Detect coastal disturbances and climate change effects in coralligenous community through  sentinel stations

PONE-D-19-25987R2

Dear Dr. Maestre,

We are pleased to inform you that your manuscript has been judged scientifically suitable for publication and will be formally accepted for publication once it complies with all outstanding technical requirements.

With kind regards,

Carlo Nike Bianchi

Academic Editor

PLOS ONE

Additional Editor Comments (optional):

Reviewers' comments:

Reviewer's Responses to Questions

**Comments to the Author**

1. If the authors have adequately addressed your comments raised in a previous round of review and you feel that this manuscript is now acceptable for publication, you may indicate that here to bypass the “Comments to the Author” section, enter your conflict of interest statement in the “Confidential to Editor” section, and submit your "Accept" recommendation.

Reviewer #2: All comments have been addressed

2. Is the manuscript technically sound, and do the data support the conclusions?

Reviewer #2: Yes

3. Has the statistical analysis been performed appropriately and rigorously? 

Reviewer #2: I Don't Know

4. Have the authors made all data underlying the findings in their manuscript fully available?

Reviewer #2: Yes

5. Is the manuscript presented in an intelligible fashion and written in standard English?

Reviewer #2: Yes

6. Review Comments to the Author

Reviewer #2: In my previous reviews I recommended major revision twice, because of the many shortcomings of the manuscript. I still have several doubts on some aspects of this manuscript including, among others, the stability of the environmental settings through the 10 years-experiment and the appropriate discussion of the outputs.

However, I find the manuscript generally improved and I agree with publication, leaving to other authors the possibility of accepting (by using it) or rejecting (by simply not using it) the method.

7. PLOS authors have the option to publish the peer review history of their article (what does this mean?). If published, this will include your full peer review and any attached files.

Reviewer #2: No

---

## [Editor Report · Acceptance letter]

24 Apr 2020

PONE-D-19-25987R2 

Detect coastal disturbances and climate change effects in coralligenous community through sentinel stations 

Dear Dr. Maestre:

I am pleased to inform you that your manuscript has been deemed suitable for publication in PLOS ONE. Congratulations! Your manuscript is now with our production department. 

With kind regards,

on behalf of

Dr. Carlo Nike Bianchi 

Academic Editor

PLOS ONE